# Differentiating Epileptic and Psychogenic Non-Epileptic Seizures Using Machine Learning Analysis of EEG Plot Images

**DOI:** 10.3390/s24092823

**Published:** 2024-04-29

**Authors:** Steven Fussner, Aidan Boyne, Albert Han, Lauren A. Nakhleh, Zulfi Haneef

**Affiliations:** 1Department of Neurology, Baylor College of Medicine, Houston, TX 77030, USA; 2Undergraduate Medical Education, Baylor College of Medicine, Houston, TX 77030, USA; 3Neurology Care Line, Michael E. DeBakey VA Medical Center, Houston, TX 77030, USA

**Keywords:** convolutional neural network, image recognition, epilepsy, electroencephalograms

## Abstract

The treatment of epilepsy, the second most common chronic neurological disorder, is often complicated by the failure of patients to respond to medication. Treatment failure with anti-seizure medications is often due to the presence of non-epileptic seizures. Distinguishing non-epileptic from epileptic seizures requires an expensive and time-consuming analysis of electroencephalograms (EEGs) recorded in an epilepsy monitoring unit. Machine learning algorithms have been used to detect seizures from EEG, typically using EEG waveform analysis. We employed an alternative approach, using a convolutional neural network (CNN) with transfer learning using MobileNetV2 to emulate the real-world visual analysis of EEG images by epileptologists. A total of 5359 EEG waveform plot images from 107 adult subjects across two epilepsy monitoring units in separate medical facilities were divided into epileptic and non-epileptic groups for training and cross-validation of the CNN. The model achieved an accuracy of 86.9% (Area Under the Curve, AUC 0.92) at the site where training data were extracted and an accuracy of 87.3% (AUC 0.94) at the other site whose data were only used for validation. This investigation demonstrates the high accuracy achievable with CNN analysis of EEG plot images and the robustness of this approach across EEG visualization software, laying the groundwork for further subclassification of seizures using similar approaches in a clinical setting.

## 1. Introduction

Epilepsy is a chronic, non-communicable disease of the brain that affects about 3.5 million people in the US [1] and 46 million worldwide [2]. Seizures in epilepsy can result in involuntary body movement, loss of consciousness, and other abnormal cognitive functions that can incapacitate a patient. Not all seizures are epileptic, however; psychogenic nonepileptic seizures (PNES) account for 25 to 35% of all epilepsy monitoring unit (EMU) discharge diagnoses [3]. The accurate distinction of PNES from epileptic seizures (ES) is critical, as the two differ in treatment and prognosis. While ES often require treatment with anti-seizure medications, implantable devices, or brain resection surgery, patients with PNES typically require psychotherapy and may be exposed to unnecessary risk if inappropriately treated for epilepsy. Distinguishing between these conditions requires referral to comprehensive epilepsy centers, and typically involves prolonged inpatient electroencephalogram (EEG) monitoring [4]. EEG monitoring uses electrodes placed on the scalp to determine the location and magnitude of electrical activity within the brain which will be altered before, during, and after an epileptic event. This process is resource-intensive and susceptible to human error, requiring an epileptologist to interpret several days’ worth of EEG signals. In fact, continuous monitoring by a qualified EEG technician or electroencephalographer is performed only in 56–80% of EMUs [5]. The development of automated seizure detection can reduce these barriers and potentially be used for seizure detection or prediction for patients outside the hospital.

Machine learning (ML) models have seen rapidly improving performance and increasing use in medicine in recent years [6]. ML models are particularly suited for the analysis of large quantities of data (“big data”) which are otherwise inefficient or impossible to process. ML bridges the fields of statistics and computer science with algorithms that “learn” based on exposure to meaningful data, rather than by explicit instructions [7]. ML approaches have been applied to various facets of epilepsy diagnosis including EEG, MRI, PET, SPECT, MEG, visual evoked potentials, and medical records analysis [8,9]. Many algorithms have been developed to detect epileptic seizures from EEG using statistical, frequency domain, nonlinear parameter, and other approaches [10,11,12,13,14,15,16].

Support vector machine (SVM) and clustering ML models have shown promise in detecting epileptic activity. Several studies demonstrate high accuracies of 90% to 99% accuracy using SVMs and clustering models trained on features extracted from pre-processed EEG signals [12,13,14,17,18,19,20]. The increasing availability of large datasets has allowed for the development of deep learning such as neural networks, which automatically learn relevant features in a supervised learning network [21]. Neural networks have also achieved excellent results using pre-processed signals, demonstrating sensitivities and specificities upwards of 90% and 99% respectively [11,22]. Convolutional neural networks (CNNs) in particular show promise for the analysis of raw EEG signals, attaining up to 98% accuracy using the time-frequency domain [23]. With respect to accuracy, both SVM and CNN models have outperformed commercially available software such as BESA^®^, Encevis, and Persyst^®^, which have a 67–81% detection rate for seizures, with a false alarm rate ranging from 0.2 to 0.9 per hour [24].

In the context of seizure diagnosis, CNN models are trained to automatically detect relevant, distinctive features of EEG images [21]. Unlike other ML analysis techniques which break down the raw electrical signals from the EEG machine, 2D CNNs interpret images by identifying key visual patterns, emulating EEG review by epileptologists in clinical practice. Recent studies have demonstrated the efficacy of CNNs for seizure classification using EEG signals, with classification accuracies reported between 82 and 98% [23,25,26,27,28,29,30]. These studies primarily focused on the identification of seizures, with one study looking at classifying seizures into different variants [25]. Previous studies have typically used an input of EEG signals preprocessed via frequency analysis techniques (e.g., wavelets, time domains, power of signal frequencies) rather than direct analysis of the visual representation of EEG waveform plot signals. Only one study has looked at visual EEG waveform plots for the identification of seizures, and reported a true positive rate of 74% with a 2D CNN trained on data from a single center [30].

Although these models perform well in training and testing, they have limitations in their generalizability due to variations in EEG patterns between patients and potentially high false alarm rates [30,31]. Furthermore, most existing models have been developed and tested using data from a single institution, and it is unclear how they would generalize given a novel dataset.

Other image-based approaches to the analysis and classification of EEG signals that do not employ CNNs have also been proposed. These approaches seek to extract information from EEG images by using a series of pre-determined mathematical operations to extract key morphological features. A notable method is the scale invariant feature transform, which identifies potential interest points and assigns descriptors through the application of image filters and gradient mapping [32,33].

In the present study, we demonstrated the efficacy of an image-based EEG waveform plot seizure classification model using CNN with transfer learning. Multi-channel EEG data from recorded clinical seizures were displayed using clinically available EEG plot review software, which were then used to train the model to classify EEG waveform images as ES or PNES. The model was then validated using data from both the original institution and from another site with a different EEG reading software.

## 2. Materials and Methods

The image-based classification of seizure types was attempted using deep learning with CNN using clinical seizure events captured at our EMU. The overview of the project is shown in Figure 1.

### 2.1. Data

EEG waveform plot images were obtained from adult subjects who were admitted to the EMU at two large academic medical centers in Houston, which are henceforth referred to as Site A and Site B. The research study was approved by the institutional review board of the Baylor College of Medicine (H-47804, H-50049).

Patient videos were divided into groups of ES and PNES and sequentially selected from patients who completed an EMU admission with at least one clinical event and a clear clinical and electrophysiologic diagnosis of ES or PNES. Videos were randomized to training, validation, and testing datasets.

During the EMU admissions, patients were monitored continuously with scalp EEG and ECG recordings and monitored by a trained neurophysiology technologist. Clinical events were either noted by push button alert by the patient, family member, or witnessed and alerted by the monitoring technologist. After completion of the EMU admission, the diagnosis of ES or PNES was determined by the supervising epileptologist based on clinical behavior of the events along with a review of the EEG waveform plot data. Subjects were excluded if they did not have any clinical events, mixed ES/PNES were present, clinical events were determined to be ES based on clinical semiology but without electrographic correlation, or if the diagnosis remained undetermined after completion of the EMU admission.

To obtain the base images, EEG waveform plots were first displayed using the clinical EEG reading software Nicolet NicVue (Natus Medical Incorporated, Pleasanton, CA, USA) and NeuroWorkbench (Nihon Kohden Corporation, Irvine, CA, USA) at Sites A and B respectively. These data were originally recorded using the 10–20 system with 2 additional electrodes (FT9/FT10). The sampling rate was 250 Hz at Site A and 1000 Hz at Site B. EEG data were filtered with a 0.3 -Hz low cut and 70 Hz high cut filters prior to data capture. Plot images are natively generated and downsampled to display 10–12 s of EEG waveform data per screen. For a 1000-pixel screen, the max frequency displayed would be 100 Hz (1000 pixels/10 s). On our display, the screenshot process resulted in a vertical resolution of approximately 1 pixel per µV.

EEG waveforms displayed on the reading software during the events of interest were captured as screenshots spanning 10–12 s epochs, which were then split into two segments of equal width representing 5–6 s epochs. The split images were then visually reviewed and split segments were excluded for the ES group if it did not contain at least 2 s of EEG data representing a clear electrographic seizure. This process ensured that all ES images contained clinically identifiable ictal activity. All PNES images, comprising a heterogenous mix of resting state EEG and the wide variety of findings in non-epileptic events, were included at this point in the dataset preparation.

EEG waveform plots in some images were not clearly visible due to dense electromyogram (EMG) artifact from the subject’s muscle activity. EEG images were therefore filtered to remove EMG-dense images from all four datasets. For this, the image array pixel values were summed up (with assigned pixel values as 0–255 for all three RGB channels) and images with summated pixel values of <18 million were removed as EMG contaminated images. The threshold of 18 million was chosen as the threshold after calculating the summated pixel value by visual inspection and manual classification of several EMG contaminated images. This was based on both our clinical judgement informed by extensive clinical experience and the success of similar methods reported in the literature [34]. Pixel data were then normalized by dividing each pixel value by 255 for all three RGB channels.

The training dataset consisted of 3061 images using four common montages used clinically at our center (AP-Bipolar, Average referential, Transverse Bipolar, and a longitudinal/transverse montage centering on FT9/FT10, henceforth called the “Phase-1” montage), with the validation and testing datasets consisting of AP-Bipolar images [Appendix A].

From Site A, there were a total of 60 adult subjects with 121 seizures (62 ES, 59 PNES) which were used for generating the training, validation, and testing datasets. Images were assigned to each dataset using a single proportional random split before beginning the training process. An additional testing dataset was collected from Site B which consisted of 47 adult subjects with 78 seizures (36 ES, and 42 PNES) for an additional out of sample evaluation using a different EEG reading software (Table 1).

### 2.2. Training and Validation Dataset

The training dataset consisted of 3061 images of 57 seizure events (33 ES, 24 PNES) and the validation dataset consisted of 518 images of 30 seizure events (14 ES, 16 PNES) at site A (Table 2).

### 2.3. Testing Dataset

The testing datasets consisted of 396 images of 34 seizure events (15 ES, 19 PNES) at Site A and 1384 images of 78 seizure events (36 ES, 42 PNES) at Site B (Table 2).

### 2.4. Convolutional Neural Network

A MobileNetV2 transfer learning model was used for CNN during pre-processing. The recommended image size for MobileNetV2 is 224 × 224 pixels. To achieve these dimensions, EEG images were converted to a square aspect ratio by image padding with a neutral background (RGB 255,0,0) and subsequently resized to 224 × 224 pixels.

The EEG classification model was trained using transfer learning using the pre-trained CNN model MobileNetV2 [35] using TensorFlow [36] and TensorFlow Hub [37]. The MobileNetV2 model architecture is composed of continuous 32 filters of convolutional layers followed by 19 residual bottleneck layers with an appended dense output layer of two nodes [35].

Transfer learning was completed by fine tuning all trainable values using a GeForce RTX 3090 (Nvidia Corp, Santa Clara, CA, USA) GPU. The stochastic gradient descent optimizer was used with a learning rate of 0.05 with a categorical cross-entropy loss function. The batch size was 32 with an image input size of 224 × 224 pixels. The model with the lowest validation loss value was saved and used for predictions.

## 3. Results

The MobileNetV2 CNN model was trained using EEG data of 3061 images from 28 subjects and validated with 518 images from 16 subjects. The output of the CNN was a classification of ‘epileptic seizure’ or ‘non-epileptic seizure’ for a given set of EEG images. The model correctly classified 90 of 138 epileptic images and 254 of 258 PNES EEG images from Site A and 390 of 408 epileptic images and 802 of 957 PNES EEG images from Site B (Table 3 and Table 4). The CNN classification results had high overall accuracies of 86.9% and 87.3% at Site A and Site B, with an area under the curve (AUC) of 0.92 for Site A and 0.94 for Site B respectively (Figure 2). Although the training data did not come from Site B, the accuracy was higher (non-significantly) than the original site (Site A). For Site A, sensitivity and specificity were 65.2% and 98.4% respectively with a positive predictive value (PPV) of 95.7% and negative predictive value of 84.1%. For Site B the sensitivity was 95.6% and specificity was 83.8% with PPV and NPV values of 71.6% and 97.8% respectively. The combined accuracy of the model on a pooled Site A and Site B testing dataset was 87.2% with a sensitivity of 87.9% and specificity of 86.9% (Table 5).

## 4. Discussion

Using transfer learning with a CNN model, we classified unknown EEG waveform plots of clinical seizure events captured at our epilepsy monitoring unit into ‘epileptic’ and ‘non-epileptic’ events. In addition to the high AUC at the original site (0.92), it was notable that the model maintained performance when tested at a different site (0.94) employing a different EEG acquisition and reviewing equipment. To the best of our knowledge, this is one of the first demonstrations of a seizure detection model maintaining high performance across multiple centers without requiring re-training to account for site-specific differences and the only such model using a CNN architecture to discriminate between epileptic seizures and PNES [24,38]. Despite only using a single random split for validation, the model was able to effectively generalize to new patients in the two test datasets, which is a consistently difficult task for many other approaches [39]. This research demonstrates the utility of visual CNN in the analysis of large EEG datasets, which was robust across different data collection/visualization technologies used in typical clinical practice.

We used a CNN to visually analyze EEG waveform plots unlike the bulk of pre-existing research which used EEG signals or spectral/wavelet analysis [23,25,26,27]. As such, our methodology more closely approximates the real-world practice of EEG review by clinicians where EEG plot images (waveforms) are reviewed during clinical analysis. Moreover, using images rather than raw signals as input to the model could allow researchers and physicians to pinpoint the specific regions of the EEG image that contribute most to the model predictions through a process known as attention mapping. This explainability offered by image-based CNN models is a critical factor for adoption in clinical practice [40].

The EEG patterns used in analysis were captured from the review software used for visual review. Like spectral/wavelet analysis, the model could have applications in the real-time distinction of epileptic vs. non-epileptic events. Moreover, our image-based algorithm requires only minimal setup such as a screen-capture program to pass data to the model. Although a screen-capture based approach to image acquisition may not provide as consistent input as direct signal-to-image conversion described in the literature [41], using screenshots allows the model to be easily implemented on top of any existing EEG reading software with minimal additional setup.

Visual EEG review is a laborious process involving hours of visual scanning by skilled electroencephalographers. While many findings are easily evident, some findings are more subtle. EEG interpretation software such as Persyst^®^ or BESA^®^ can reduce physician workload, but the review process still requires considerable effort. There is one previous study which evaluated visual CNNs to classify EEG plot images as “seizures” and “non-seizures” [30]. They found that the true positives with the visual CNN method by one-second windows (74%) outperformed the established commercial EEG detection software—BESA^®^ (20%) and Persyst^®^ (31%). Using one-minute windows, the previous work achieved a detected seizure rate of 100% compared to 73.3% for BESA^®^ and 81.7% for Persyst^®^. Our analysis similarly suggests CNNs are superior to commercially available seizure detection software, and we further improve upon the seizure classification accuracy from the previous work. In comparison with the previous study which analyzed 1-s EEG segments, we analyzed 6-s segments, which may have contributed to the improved performance and more closely approximates the duration of EEG analyzed at a time by human experts (typically 10–15 s EEG segments per screen). When comparing the performance of these models to the current work, however, it is important to note commercial EEG software is designed to detect seizure from baseline activity, not distinguish epileptic activity from PNES.

Only a select few machine learning methods have been developed with the goal of sub-classifying seizure variants. Several ML models have been trained to distinguish PNES from epileptic seizure, achieving accuracies ranging from 87% to 95% (BayesNet 95%, Random Committee 89%, and Random Forest 87%) with an input of several pre-selected EEG features [42]. CNN models have even been trained to sub-classify epileptic events into categories such as absence seizure, focal seizures, and grand-mal seizures, reaching up to 88.3% accuracy [25,38]. On the other hand, there is a wealth of literature that uses machine learning to analyze raw EEG signals, although these studies focus on detecting rather than classifying seizures (see [15,16,43] for reviews).

In the current study, we look at EEGs from clinical events rather than ictal-interictal EEG to classify them as “epileptic” and “non-epileptic” seizures. PNES are a heterogeneous group of behavioral episodes where the patient develops a seizure-like event in response to a presumed stressor. These can take the form of altered behavior including staring, non-responsiveness, nonsensical speech, and abnormal movements including body movements, shaking, back arching, and side-to-side head movements. These are not generated by abnormally synchronized electrical activity in the brain as occurs with seizures. As such, the EEG pattern does not show a change from baseline, although there are often artifacts in the EEG due to abnormal movements, which can sometimes be difficult to distinguish from seizures.

The use of AUC of the receiver operating curve has become widely accepted as a standardized way of reporting classification accuracy [44]. A head-to-head comparison of different signal analysis methods in mouse models revealed the superiority of CNNs (AUC 0.989–0.993) compared to fully connected neural networks (AUC 0.983–0.984) and recurrent neural networks (AUC 0.985–0.989) in classifying epileptic from interictal EEG in raw intracranial EEG waveforms [45]. Recent studies have shown comparable performance in humans with an AUC of 0.83 using CNN’s for detecting interictal epileptiform discharges [29]. In our study, we show the ability of CNNs to produce similar AUCs of 0.92–0.94 in a clinical setting with human patients when classifying EEG waveform plots from epileptic and non-epileptic clinical events.

## 5. Conclusions

Currently, review of EEG for purposes of seizure classification is laborious and requires time-intensive effort by skilled electroencephalographers. The use of real-time EEG analysis by classification software would reduce effort and potential error by human observers. Additionally, this facilitates continuous monitoring, faster diagnosis, and reduction in medical costs. We demonstrate high accuracy in the distinction of non-epileptic from epileptic seizures, which was robust across sites and different review software, indicating that such a technique could have good real-world performance. The accuracy could potentially be improved with larger datasets and the additional fine-tuning of CNN parameters. Areas for future work include the exploration of attention mapping for a more explainable model and investigation into the use of vision based methods on resting state EEG which has shown promise as a more easily obtainable source of diagnostic information [40,46].

## Figures and Tables

**Figure 1 sensors-24-02823-f001:**
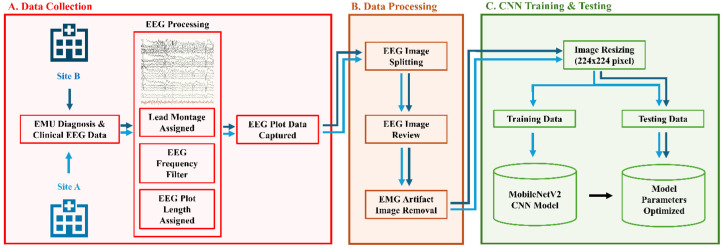
Project Overview (**A**) Clinical EEG data obtained from EMU sites after epileptic/non-epileptic seizure identified from Site A (light blue arrow) and Site B (dark blue arrow). EEG waveforms processed with frequency filter, lead montage, and plot length of 10–12 s epochs before image capture of EEG. (**B**) Images were processed by splitting into 2 segments of 5–6 s epochs. Images were visually reviewed and filtered through an EMG artifact program to remove images without electrographic seizure or excess EMG artifact. (**C**) Images were resized and split into the training dataset from Site A and testing dataset from Site A and B, with Site B providing out-of-sample evaluation with different EEG software. Images input into MobileNetV2 CNN model.

**Figure 2 sensors-24-02823-f002:**
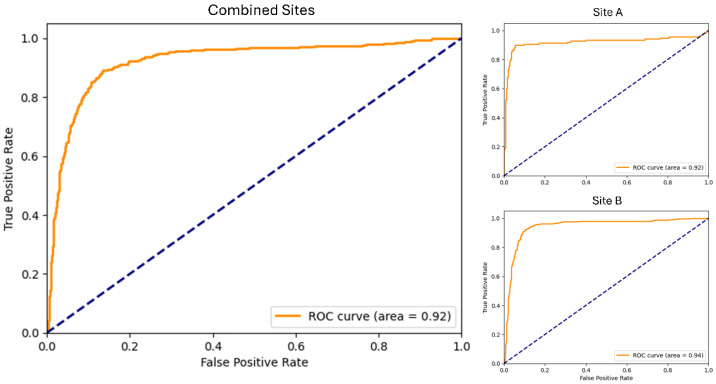
Area under the receiver operating curve (AUC) of the model on the combined validation images, Site A validation images, and Site B validation images.

**Table 1 sensors-24-02823-t001:** Subject Age by Dataset.

	Site A—Training Dataset		Site A—Validation Dataset		Site A—Testing Dataset		Site B—Testing Dataset	
	Count	Age (mean ± SD)	Count	Age (mean ± SD)	Count	Age (mean ± SD)	Count	Age (mean ± SD)
ES	15	51 ± 14	8	44 ± 14	7	49 ± 16	22	39 ± 12
PNES	13	48 ± 12	8	36 ± 13	9	49 ± 14	25	41 ± 14
Total	28	50 ± 13	16	40 ± 13	16	49 ± 14	47	40 ± 13

ES—epileptic seizure, PNES—psychogenic non-epileptic seizure, SD—standard deviation.

**Table 2 sensors-24-02823-t002:** Details of image plots used for training, validation and testing at Site A and B.

Image Usage	Source	Number of Subjects	ES	ES Images	PNES	PNES Images	Total Images
Training	Site A	28	33	1438	24	1623	3061
Validation	Site A	16	14	228	16	290	518
Testing 1	Site A	16	15	138	19	258	396
Testing 2	Site B	47	36	408	42	957	1365

ES—epileptic seizure, PNES—psychogenic non-epileptic seizure.

**Table 3 sensors-24-02823-t003:** Model testing results from Site A.

	Seizure Type		Total Images
Model Classification	Epileptic	Non-Epileptic	
Epileptic	90	4	94
Non-Epileptic	48	254	302
Total	138	258	396

**Table 4 sensors-24-02823-t004:** Model testing results from Site B.

	Seizure Type		Total Images
Model Classification	Epileptic	Non-Epileptic	
Epileptic	390	155	545
Non-Epileptic	18	802	820
Total	408	957	1365

**Table 5 sensors-24-02823-t005:** Model results for Site A, Site B, and combined analysis.

Measure	Site A (95% CI)	Site B (95% CI)	Combined Sites (95% CI)
Sensitivity	65.2% [57.0% to 72.7%]	95.6% [93.1% to 97.2%]	87.9% [84.9% to 90.4%]
Specificity	98.4% [96.1% to 99.4%]	83.8% [81.3% to 86.0%]	86.9% [84.9% to 88.9%]
PPV	95.7% [89.6% to 98.3%]	71.6% [67.6% to 75.2%]	75.1% [71.6% to 78.3%]
NPV	84.1% [79.6% to 88.0%]	97.8% [96.6% to 98.6%]	94.1% [92.6% to 95.3%]
Accuracy	86.9% [83.2% to 89.8%]	87.3% [85.5% to 89.0%]	87.2% [85.6% to 88.7%]

PPV—positive predictive value, NPV—negative predictive value. CI—confidence interval. Confidence intervals for each metric were calculated using the Wilson score interval.

## Data Availability

The data that support the findings of this study are available on request from the corresponding author, Z.H., subject to a data use agreement with the institutions at which the study was conducted. The data are not publicly available due to institutional regulations.

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
