# Peer review of "Differentiating Epileptic and Psychogenic Non-Epileptic Seizures Using Machine Learning Analysis of EEG Plot Images"

_sensors, 2024, doi:10.3390/s24092823_

Round 1
Reviewer 1 Report
Comments and Suggestions for Authors
Dear Authors,
The work presented here is a very interesting approach to implement an intrinsically explainable method to detect patterns in EEG signals, with the important application of epileptic seizure detection. I believe this is very important in terms of offering a new approach to solve the issue of epileptic seizure detection and classification from a clinically relevant perspective.
However, I have found some important issues that require careful consideration.
1) There is a lack of proper citation of similar works that are based on the same idea. A description of the previous work should be advisable to understand differences and to highlight the important contribution of this particular work. I would suggest to check the work published https://ieeexplore.ieee.org/abstract/document/7457454. Many of the ideas described in this work are also expounded in these previous works, albeit not for seizure detection itself (which I think it is the contribution of this work) (lines 70-82 of the manuscript, 228-234).
2) There is a lack of more theoretical understanding of what it really means to convert a signal to an image that is going to be used as input of a neural network. A baseline theoretical model is described in chapter 3 of this thesis https://www.researchgate.net/publication/329327990_Histogram_of_Gradient_Orientations_of_EEG_Signal_Plots_for_Brain_Computer_Interfaces. This is truly important for reproduction of the method, and should be described in detail. For instance, there isn't any description of how the resolution on the vertical axis is being performed to convert to uV to pixels (lines 117-125). Additionally, it is not clear in line 140 how the conversion is being performed between amplitud values and pixel values, or the connection between channels (from EEG) to colors in images.
3) This manuscript should compare the solution presented here with other SOTA methods on this dataset, and also to check the applicability of this method in other standard epilepsy datasets (at least one). For instance, other standard CNN based methods which are commonly used for EEG, like EEGNET or BCINET and to check the performance that could be obtained with those methods. The point here is to understand if the difference in performance is not significant to justify using a method that is explainable.
4) I believe authors are missing an important point which the intrinsically explainability offered by this approach in the area of XAI (Explainable Artificial Intelligence).
5) It will be interested if it is possible for the Authors to publish the dataset that has been created. This will help others to verify and test algorithms and compare them with the method proposed in this manuscript.
Author Response
We thank the reviewer for their feedback. The comments and suggestions raise many important points which have strengthened the manuscript. Below, we discuss how we addressed each concern in red, list any changes to the text in blue, and detail where the accompanying revisions may be found in the revised manuscript.
Dear Authors,
The work presented here is a very interesting approach to implement an intrinsically explainable method to detect patterns in EEG signals, with the important application of epileptic seizure detection. I believe this is very important in terms of offering a new approach to solve the issue of epileptic seizure detection and classification from a clinically relevant perspective.
However, I have found some important issues that require careful consideration.
1) There is a lack of proper citation of similar works that are based on the same idea. A description of the previous work should be advisable to understand differences and to highlight the important contribution of this particular work. I would suggest to check the work published https://ieeexplore.ieee.org/abstract/document/7457454. Many of the ideas described in this work are also expounded in these previous works, albeit not for seizure detection itself (which I think it is the contribution of this work) (lines 70-82 of the manuscript, 228-234).
Response: A further discussion of imaged-based EEG analysis techniques beyond the ML methods discussed in lines 57-82 is now included in the introduction (lines 88-93). The study recommended along with one other source discussing image-based EEG analysis are cited (references 32 and 33).
88-93: Other image-based approaches to the analysis and classification of EEG signals that do not employ CNNs have also been proposed. These approaches seek to extract information from EEG images by using a series of pre-determined mathematical operations to extract key morphological features. A notable method is the scale invariant feature transform, which identifies potential interest points and assigns descriptors through the application of image filters and gradient mapping [32,33].
2) There is a lack of more theoretical understanding of what it really means to convert a signal to an image that is going to be used as input of a neural network. A baseline theoretical model is described in chapter 3 of this thesis https://www.researchgate.net/publication/329327990_Histogram_of_Gradient_Orientations_of_EEG_Signal_Plots_for_Brain_Computer_Interfaces. This is truly important for reproduction of the method, and should be described in detail. For instance, there isn't any description of how the resolution on the vertical axis is being performed to convert to uV to pixels (lines 117-125). Additionally, it is not clear in line 140 how the conversion is being performed between amplitude values and pixel values, or the connection between channels (from EEG) to colors in images.
Response: The reviewer’s point is well taken. An expanded discussion of the signal to image conversion processes is added to the methods section of the manuscript (lines 131-132, 134, 150-151) with reference to the thesis mentioned above (reference 41). However, it is important to note that the lack of standardized signal to image conversion (such as the method described in the referenced thesis by R. Ramele et al.) is intentional, and a key contribution of this work. Though such processes may provide a more consistent input more easily and accurately classified by CNN models, we sought to make our approach as portable and generalizable as possible by training the model with simple screenshots of the workbench images. By using screenshots rather than pre-processed signal converted to images, input for the model can easily be collected at any institution using any standard EEG reading software and fed directly to the model without the need for import and conversion of the raw signal data. This rationale is now explained further in the discussion (lines 256-259).
131-132: On our display, the screenshot process resulted in a vertical resolution of approximately 1 pixel per µV…
134: EEG waveforms displayed on the reading software during the events of interest were captured as screenshots spanning 10-12 second epochs
150-151: Pixel data were then normalized by dividing each pixel value by 255 for all three RGB channels.
3) This manuscript should compare the solution presented here with other SOTA methods on this dataset, and also to check the applicability of this method in other standard epilepsy datasets (at least one). For instance, other standard CNN based methods which are commonly used for EEG, like EEGNET or BCINET and to check the performance that could be obtained with those methods. The point here is to understand if the difference in performance is not significant to justify using a method that is explainable.
Response: We searched standard epilepsy datasets listed in a recent review by Wong et al. and found no dataset with both PNES and ES activitya. For example, two of the more prominent standard datasets Bonn and CHB-MIT contain only control and epilepsy data without any PNES data. Furthermore, our dataset consists of screenshots taken from the EEG reading software, not the raw signal data. It would not be feasible to obtain the raw signal data required for comparison EEGNET or BCINET. We discuss the performance of our model against several CNN based methods in lines 62-82, 265-274, 278-286, and 297-308, though we do realize the limitations in this comparison due to the different datasets used for testing and training. These limitations are now elaborated in the discussion section (lines 274-277).
a. Wong, S.; Simmons, A.; Rivera‐Villicana, J.; Barnett, S.; Sivathamboo, S.; Perucca, P.; Ge, Z.; Kwan, P.; Kuhlmann, L.; Vasa, R.; et al. EEG Datasets for Seizure Detection and Prediction— A Review. Epilepsia Open 2023, 8, 252–267, doi:10.1002/epi4.12704.
274-277: When comparing the performance of these models to the current work, however, it is important to note commercial EEG software is designed to detect seizure from baseline activity, not distinguish epileptic activity from PNES.
4) I believe authors are missing an important point which the intrinsically explainability offered by this approach in the area of XAI (Explainable Artificial Intelligence).
Response: This is an excellent point – explainability is a key advantage of vision-based models that we neglected to discuss in sufficient detail. The interpretability of the model and why explainable AI is important in the field are now discussed more thoroughly (lines 247-251, 318-319).
247-251: Moreover, using image rather than raw signal as input to the model could allow researchers and physicians to pinpoint the specific regions of the EEG image that contribute most to the model predictions through a process known as attention mapping. This explainability offered by image-based CNN models is a critical factor for adoption in clinical practice [40].
318-319: Areas for future work include the exploration of attention mapping for a more explainable model… [40,46].
5) It will be interested if it is possible for the Authors to publish the dataset that has been created. This will help others to verify and test algorithms and compare them with the method proposed in this manuscript.
Response: Unfortunately, the dataset used to conduct this study was collected under an IRB protocol which did not include provisions for data sharing. The institutions at which the study was preformed require special clearances for data sharing. We have updated the data availability statement to reflect this and guide researchers who seek to replicate the study to reach out for more details.
Data Availability Statement: The data that support the findings of this study are available on request from the corresponding author, Z.H., subject to a data use agreement with the institutions at which the study was conducted. The data are not publicly available due to institutional regulations.
Reviewer 2 Report
Comments and Suggestions for Authors
The quest for an automatic method to detect PNES patients vs. epileptic patients is clinically relevant. The methodology used in the manuscript is interesting. However, I believe that some points need to be addressed before publication.
What kind of cross-validation the authors use? A recent methodological work, namely Shafiezadeh et al., 2023 -Methodological Issues in Evaluating Machine Learning Models for EEG Seizure Prediction: Good Cross-Validation Accuracy Does Not Guarantee Generalization to New Patients, evidenced how the classification percentage in epilepsy drops when leave-one out cross-validation is applied. Please provide information about the cross validation approach used in this classification and include this reference in the manuscript.
It is not clear to me if the authors selected interictal ictal EEG (presumed ictal in the context of PNES) for the classification. In case they used interical EEG signal, can the authors please specify the distance from the ictal event? In fact, the classification using interictal EEG may be biased as a function of distance from the ictal event.
In relation to this point a possible solution would be to use resting state EEG, for example comparing patients with epilepsy diagnosis vs. PNES. In this way one can be sure to investigate the proper brain dynamics characterizing epilepsy. Recent works in fact used resting state EEG without interictal spikes to detect functionally altered brain regions in epilepsy and to perform automatic classification.
Duma et al., 2023 - Altered spreading of neuronal avalanches in temporal lobe epilepsy relates to cognitive performance: A resting‐state hdEEG study, Epilepsia
I believe the authors should add a paragraph in the discussion about the generalizability of their method in easy clinical scenario, like 10 minutes resting state EEG recording, including at least the above mentioned reference.
Author Response
We thank the reviewer for their feedback. The comments and suggestions raise many important points which have strengthened the manuscript. Below, we discuss how we addressed each concern in red, list any changes to the text in blue, and detail where the accompanying revisions may be found in the revised manuscript.
The quest for an automatic method to detect PNES patients vs. epileptic patients is clinically relevant. The methodology used in the manuscript is interesting. However, I believe that some points need to be addressed before publication.
What kind of cross-validation the authors use? A recent methodological work, namely Shafiezadeh et al., 2023 -Methodological Issues in Evaluating Machine Learning Models for EEG Seizure Prediction: Good Cross-Validation Accuracy Does Not Guarantee Generalization to New Patients, evidenced how the classification percentage in epilepsy drops when leave-one out cross-validation is applied. Please provide information about the cross validation approach used in this classification and include this reference in the manuscript.
Response: The referenced work makes an excellent point, and the drop in performance statistics with leave-one out cross-validation is indeed striking. We used a single proportional random split to divide our data into training, validation, and testing sets. We have added sections discussing both our validation approach in the methods section (lines 158-160) and the potential limitations of such an approach in the discussion section (lines 229-231) as outlined by the work by Shafiezadeh et al. The high performance of our model on the test datasets from Site A and Site B (both of which consisting entirely of images not previously analyzed by the model, and site B consisting of events from patients not included in the training process and displayed using a different EEG reading software) suggests that the model successfully generalized to new patients despite the use of a simple data stratification strategy.
158-160: Images were assigned to each dataset using a single proportional random split before beginning the training process.
229-231: Despite only using a single random split for validation, the model was able to effectively generalize to new patients in the two test datasets, a consistently difficult task for many other approaches.
It is not clear to me if the authors selected interictal ictal EEG (presumed ictal in the context of PNES) for the classification. In case they used interical EEG signal, can the authors please specify the distance from the ictal event? In fact, the classification using interictal EEG may be biased as a function of distance from the ictal event.
Response: In the case of the epileptic seizure (ES) group, each 5-6 second epoch contains at least 2 seconds of ictal activity as stated in lines 133-137. Accordingly, there are no clips consisting exclusively of interictal or pre-ictal activity in the ES group, and the maximum time from interictal or pre-ictal activity to an ictal event is 4 seconds. We further clarified this point in the text (lines 137-140) as it is indeed an important methodological decision. The images for the PNES group are a heterogeneous mix of normal background activity and the wide variety of findings seen with NES events.
133-137: EEG waveforms displayed on the reading software during the events of interest were captured as screenshots spanning 10-12 second epochs, which were then split into two segments of equal width representing 5-6 second epochs. The split images were then visually reviewed and split segments were excluded for the ES group if it did not contain at least 2 seconds of EEG data representing a clear electrographic seizure.
137-140: This process ensured that all ES images contained clinically identifiable ictal activity. All PNES images, comprised of a heterogenous mix of resting state EEG and the wide variety of findings in non-epileptic events, were included at this point in the dataset preparation.
In relation to this point a possible solution would be to use resting state EEG, for example comparing patients with epilepsy diagnosis vs. PNES. In this way one can be sure to investigate the proper brain dynamics characterizing epilepsy. Recent works in fact used resting state EEG without interictal spikes to detect functionally altered brain regions in epilepsy and to perform automatic classification.
Duma et al., 2023 - Altered spreading of neuronal avalanches in temporal lobe epilepsy relates to cognitive performance: A resting‐state hdEEG study, Epilepsia
I believe the authors should add a paragraph in the discussion about the generalizability of their method in easy clinical scenario, like 10 minutes resting state EEG recording, including at least the above mentioned reference.
Response: The work by Duma et al. is a very interesting approach to seizure classification and certainly holds potential for the differentiation of epileptic seizures from PNES. Although our model is not trained to detect the potential differences in resting state EEG, Duma et al. highlight what seems to be an underutilized source of data for seizure classification and we have added comment in the conclusion concerning implications for future research and cited the work (lines 318-321, reference 46).
318-321: Areas for future work include the exploration of attention mapping for a more explainable model and investigation into the use of vision based methods on resting state EEG which has shown promise as a more easily obtainable source of diagnostic information [40,46].
Round 2
Reviewer 2 Report
Comments and Suggestions for Authors
the authors replied to my comments